# Emerging Effects of Resveratrol on Wound Healing: A Comprehensive Review

**DOI:** 10.3390/molecules27196736

**Published:** 2022-10-09

**Authors:** Yuan Jia, Jia-Hao Shao, Kai-Wen Zhang, Ming-Li Zou, Ying-Ying Teng, Fan Tian, Meng-Nan Chen, Wei-Wei Chen, Zheng-Dong Yuan, Jun-Jie Wu, Feng-Lai Yuan

**Affiliations:** 1Wuxi Clinical Medicine School of Integrated Chinese and Western Medicine, Nanjing University of Chinese Medicine, Wuxi 214041, China; 2Wuxi Clinical Medicine Hospital of Chinese Medicine, Nanjing University of Chinese Medicine, Wuxi 214041, China; 3Department of Burns and Plastic Surgery, the Affiliated Hospital of Jiangnan University, Wuxi 214041, China

**Keywords:** resveratrol, wound healing, wound dressing, anti-inflammatory, anti-oxidant

## Abstract

Resveratrol (RSV) is a natural extract that has been extensively studied for its significant anti-inflammatory and antioxidant effects, which are closely associated with a variety of injurious diseases and even cosmetic medicine. In this review, we have researched and summarized the role of resveratrol and its different forms of action in wound healing, exploring its role and mechanisms in promoting wound healing through different modes of action such as hydrogels, fibrous scaffolds and parallel ratio medical devices with their anti-inflammatory, antioxidant, antibacterial and anti-ageing properties and functions in various cells that may play a role in wound healing. This will provide a direction for further understanding of the mechanism of action of resveratrol in wound healing for future research.

## 1. Introduction

The skin is the first line of defence against the external environment, which is divided into an epidermis and a dermis [1]. When external physical damage and chemical stimuli reach the deep dermis, skin trauma repair time is prolonged. During the repair process, a number of cytokines are activated by the immune response and oxidative stress of the body [2]. However, excessive activation of these cytokines eventually leads to the development of pathological scarring, such as hyperplastic scars and keloids. In addition, the healing of diabetic wounds has long been a clinical challenge [3,4,5], which has a significantly negative impact on people’s mental and social lives. For this reason, natural compounds with anti-inflammatory, antioxidant and antibacterial properties have been identified for wound healing.

Resveratrol (RSV) is a naturally occurring non-flavonoid polyphenolic compound derived from plants and it belongs to the class of stilbene compounds [6,7]. It is a colourless needle-like crystal, insoluble in water and soluble in ether, chloroform and alcohol solvents. It is mainly found in red wine, spruce bark, rice, grape skins, peanuts and berries [8] and has been shown to have antioxidant, anti-inflammatory, anti-ageing, anti-diabetic and pro-angiogenic regulatory effects with a wide range of physiopathological effects. Compared to the use of antibiotics, RSV has the advantage of being of natural origin and low cytotoxicity. However, due to its rapid drug metabolism and excretion, the drug is rapidly degraded when administered orally and its bioavailability is poor [9,10]. In order to improve RSV drug utilization, various forms of RSV-loaded dressings have been developed and are widely used in the pharmaceutical, cosmetic and nutraceutical industries.

This review expounds that the role of RSV and its application dressings in wound healing. We described the effects of RSV on different effector cells in the process of wound healing and the underlying mechanisms of RSV action were summarized, containing anti-inflammatory, antibacterial, antioxidant and anti-aging effects. It provides a reference for further research on the pharmacological effects of RSV in wound healing.

## 2. The Performance Forms of RSV to Wound Healing

Due to the low bioavailability of RSV for oral ingestion and the water-insoluble nature of RSV, various types of wound dressings have been developed for topical treatment [11,12]. Increasingly, studies have designed new drug delivery systems for topical application of RSV or have modified the structural characteristics of RSV to improve its bioavailability during systemic ingestion and eventually develop this potential molecule as a first-line treatment against various diseases [13]. However, before the clinical potential of RSV can be fully accepted, appropriate application protocols must be designed for both systemic and topical subcutaneous. RSV can also be incorporated into different types of wound dressings, such as hydrogels [14] and biological scaffolds [15], to preserve its biological function and to integrate with the surrounding host tissue. Figure 1 shows the basic form of RSV application in wound healing. Biomaterials used in skin delivery systems must meet the requirements of biocompatibility, biodegradability, good moisture and oxygen permeability, good adhesion and permeability, and reduced infection and mechanical irritation.

RSV is a natural compound derived from peanut, grape peel and spruce bark. Currently, the application of RSV is divided into natural compound and loaded dressing, which includes hydrogels, scaffolds, Dermalix, nanovesicles and wafer.

### 2.1. Hydrogel Chitosan-Based Wound Dressing

Hydrogels are a polymer with a three-dimensional network structure formed by cross-linking hydrophilic, which can absorb large amounts of water and not be dissolved by water. Hydrogels can absorb large amounts of water and swell under surface tension and capillary action, where the three-dimensional network structure prevents breakage or dissolution during the swelling process [16]. Smart hydrogels are hydrogels that are responsive to temperature, pH, electric field and light, etc. These hydrogels have unique physicochemical properties, bio-softness and compatibility, tissue adhesion and injectability. Chitosan, a derivative of chitin, is a natural alkaline cationic polysaccharide second only to cellulose in nature, with good water absorption and moisturizing properties, biocompatibility and bio-degradability [17]. Studies have concluded that chitosan has antibacterial effects on most bacteria and has low cytotoxicity [18]. Chitosan dissolves in weakly acidic solutions and generates protonated amine (-NH^3+^), which can combine with negatively charged macromolecules on the surface of bacteria, forming a dense film on the surface of bacteria, affecting the exchange of substances inside and outside the bacteria and inhibiting the growth and reproduction of bacteria [19]. Therefore, chitosan hydrogels are widely used in biological fields such as drug delivery, biomedicine and tissue engineering. Mihai-Stefan Muresan et al. discovered that a polymer-based sponge containing chitosan-sodium hyaluronate-RSV was formed by titrating the chitosan gel with 10% sodium hydroxide at a stable pH of 9, causing cross-linking of the chitosan gel and the hyaluronic acid gel, followed by mixing the RSV solution gel. This gel dressing has good biocompatibility and tissue adhesion, and has certain antibacterial and anti-inflammatory effects when combined with RSV drugs [20]. The combination of RSV-containing-acrylic-modified cyclodextrin and gelatin molecules through host–guest assembly results in a hydrogel with good shear and injectable properties for easier injection and rapid adaptation to irregular wounds [21]. In addition, RSV-encapsulated hydrogels can be prepared by freezing and thawing techniques to form RSV-loaded PVA Cryogel Membranes. These Cryogel Membranes are more effective in spreading RSV to the wound with their good mechanical and elastic properties [22]. In addition to direct complexation with bioactive substances, RSV can be used as carriers for the preparation of structures to improve their structural physical properties, such as scaffolds.

### 2.2. Electrospun Scaffolds

In trauma treatment, excellent trauma dressings can significantly reduce the rate of infection, disability and death. Traditional dressings such as gauze and bandages have many defects, such as poor moisturizing and anti-infection effects, and the trauma surface tends to adhere to the dressing, resulting in pain and damage to new tissues during dressing changes. With the development of medical technology in recent years, various new trauma dressings have been increasingly researched and applied. Electrospun scaffolds are a technical method to prepare nanofibers, which are simple and low cost compared with traditional spinning methods [23]. Scaffolds have unique size advantages, which can mimic the structure and function of biological extracellular matrix (ECM), promoting cell proliferation and accelerating wound healing, and show great potential in the field of medical dressings. Electrostatic poly(ɛ-caprolactone) (PCL) spinning scaffolds were prepared by using PCL as a co-solvent in a 7:3 ratio of chloroform and dimethylformamide, and 5% RSV solution 5% (*w/v*) RSV solution was added to form RSV-loaded electrostatic spinning scaffolds to accelerate wound closure and re-epithelialization [23]. In addition, the bilayer scaffolds formed by the combination of RSV-loaded electrostatic spun silk scaffolds and hydrogels have good haemostatic and swelling reduction abilities in wound healing [24].

### 2.3. Others

Besides the more classic hydrogels and scaffolds, a number of new RSV-loaded adjunctive therapies have been explored, including Dermalix, nanovesicles, wafers. Dermalix [25] is a microparticle formed by the fibrous component collagen, the cell-binding protein laminar flow adhesion protein bound to the skin matrix and hyaluronic acid, and the cell membrane lipid dipalmitoyl phosphatidylcholine, which have good mechanical stability and permeability, providing mechanical support for the effective release of RSV into the wound environment and improving wound healing efficiency. Wafers have been extensively studied as wound dressings. Compared to semi-solid polymer gels that flow easily, wafers sheets can maintain a swollen gel structure for long periods of time. It is a solid dosage form with a highly porous structure prepared by freeze-drying of the gel or polymer solution, which has good drug loading and water absorption capacity. The combination of RSV-loaded nanoparticles with the wafer increases the residence time of the drug in the skin [26]. In addition, the combination of RSV and gallic acid, two natural phenols with different water solubility, with phospholipids, forms a single vesicle with antioxidant and antibacterial activity, protecting the skin from external contamination and damage [27]. Binding of RSV-carrying HA-DPPC particles to a collagen–laminin dermal matrix by using the physicist method enhances the stability of RSV release and improves diabetic wound healing efficacy with its antioxidant properties [28].

## 3. Beneficial Effects of RSV on Different Cells

Wound healing is an evolutionarily conserved, complex, multicellular process that involves the synergistic action of multiple cells, including endothelial cells, fibroblasts, keratinocytes, macrophages and mesenchymal stem cells [29]. The migration, infiltration, proliferation, and differentiation of these cells play key roles in various stages of the inflammatory response, granulation tissue formatio and tissue remodelling. Figure 2 shows the beneficial cells of RSV effects on wound healing. These cells are recruited at various stages of wound healing and contribute to tissue repair and remodelling.

Skin is mainly divided into epidermis, dermis and subcutaneous tissue. There are many cells involved in the wound healing process, including keratinocytes, fibroblasts, macrophages and mesenchymal stem cells.

### 3.1. Fibroblasts

Fibroblasts are the most abundant cell type in the dermis and the one that plays the most crucial role in wound healing. Wound healing is a dynamic process with four main phases: haemostasis, inflammation, granulation formation and tissue remodelling. The movement of fibroblasts extends from the inflammatory phase to the tissue remodelling phase of wound healing. In general, fibroblasts can be collectively referred to as cells that express the genes encoding collagen I alpha chain (COL1A), platelet-derived growth factor receptor-alpha (PDGFRα) in the resting state [30]. Some reports suggest that dermal fibroblasts can promote wound healing from the late inflammatory phase by the secretion of growth factors, cytokines, collagen and other extracellular matrix (ECM) components until complete re-epithelialization of the injured tissue [31]. Given the central role of fibroblasts in wound healing, the alterations in the expression and secretion of a number of collagen factors are frequently found, including in hypertrophic scars and keloids. Fibroblasts with high expression of α-SMA, Col-1, Col-3 are more frequent in scars compared to non-injured tissue, characterizing by increased proliferation migration of fibroblasts and even differentiation to myofibroblasts, leading to excessive accumulation of collagen fibres and deposition of ECM. The process of wound injury is inseparable from oxidative stress. Under sustained oxidative stress conditions, low doses of RSV were able to reverse the H_2_O_2_-induced reduction in fibroblast proliferation and migration and accelerate wound healing by stabilizing the ultrastructure of fibroblasts and promoting collagen fibre alignment during the antioxidant process. However, the presence of autophagic vacuoles was also found in the RSV group, suggesting that there are still shortcomings in the treatment of wound healing by RSV [32]. When it comes to the effect of autophagy in wound healing, some researchers also found that fibroblasts exposed to 100uM/L RSV inhibited their binding with Rheb by stimulating miR-4654 to induce autophagy and apoptosis, ultimately inhibiting the formation of hypertrophic scars [33]. The above suggests that RSV has a dose-dependent effect on fibroblast proliferation and apoptosis in wound healing. In addition, RSV can also regulate the proliferation and apoptosis of pimple fibroblasts by targeting HIF-1α [34].

The TGF-β1/Smads signalling pathway is the most classical mechanism of action in wound healing. Several studies have suggested that high doses of RSV can induce apoptosis and inhibit the proliferation of scar fibroblasts. RSV inhibits fibroblast proliferation and induces apoptosis through downregulation of TGF-β1 and Smad2, Smad3 protein and gene level expression [35], which can be regarded as a pathway for the treatment of pathological scarring. Further studies have found that RSV inhibits the activation of TGF-β1 on normal fibroblasts through upregulation of Sirtuin1 (SIRT1), suggesting that SIRT1 can act as an upstream target of TGF-β to reduce the formation of proliferative scarring [36]. Aside from TGF-β1/Smads, downregulation of the mammalian target of (mTOR)/70S6K signalling pathway has also been investigated as a mechanism to inhibit scarring fibroblasts [37].

### 3.2. Keratinocytes

Keratinocytes are the main constituent cells of the skin’s epidermis that are resistant to microorganisms and prevent ultraviolet light from entering the deeper layers of the skin. It is thought that keratinocytes can be used as a skin substitute for wound treatment, and the proliferation and migration of keratinocytes contribute to the healing of burn wounds. During wound healing, the secretion of keratinocyte growth factor is one of the important factors in promoting re-epithelialization, which promotes the migration and proliferation of keratinocytes towards the wound edge to restore the epidermal barrier in the wound bed [38]. In addition, it activates and promotes the proliferation of downstream cellular fibroblasts with a paracrine manner [39]. In a lipopolysaccharide-induced human epidermal keratinocytes (HaCaT) injury model, RSV promoted skin wound recovery by targeting the miR-212/CASP8 axis of healing [40]. In particulate matter (PM)-induced HaCaT, RSV reduced PM-induced expression of cyclooxygenase-2 (COX-2)/prostaglandin E2 (PGE2) and pro-inflammatory cytokines, including matrix metalloproteinase (MMP)-1, MMP-9 and interleukin-8 (IL-8), to promote skin injury repair with anti-inflammatory and antioxidant properties [41]. Interestingly, in H_2_O_2_-induced HaCaT [42], grape seed proanthocyanidin extract (GSPE) containing 5000 ppm RSV accelerated wound contraction and closure by driving VEGF to promote angiogenesis, indirectly demonstrating RSV’s ability to treat traumatic injuries. In the process of wound healing, SIRT1 acts not only on fibroblast proliferation but also stimulates the growth of keratinocytes. It was found [43] that class III and class I histone deacetylases interacted with each other in a NO-dependent manner without the activation of SIRT1, stimulating keratinocytes proliferation through phosphorylation of endothelial NO synthase and NO production.

In conclusion, the above suggests that RSV promotes keratinocytes proliferation and accelerates wound repair and healing via its anti-inflammatory, antioxidant and pro-angiogenic properties. Surprisingly, Liudmila G. Korkina et al. [44] believed that RSV blocked the cell cycle via the EGFR/P-EGFR axis and inhibited keratinocytes proliferation, which was detrimental to wound healing. Therefore, considering the dose-dependent nature of RSV, its action on keratinocytes repair still needs to be further explored.

### 3.3. Macrophages

Macrophages are important inflammatory cells that play a key role in fighting infection in the body during wound healing. However, it has been suggested that an increase in excessive inflammatory cytokines is positively associated with the severity of trauma [45]. Therefore, balancing macrophages homeostasis is also a critical step in wound healing. During the early stages of wound healing, macrophages produce several pro-inflammatory cytokines such as IL-6, IL-1β and tumour necrosis factor-α (TNF-α) [46]. RSV-loaded hydrogels reduced macrophages infiltration and the expression of cytokines such as TNF-α and IL-1β in a rat skin damage model, which promoted shrinkage and repair of wounds [47]. Similarly, lipopolysaccharide-induced macrophages activation exhibited a marked increase in inflammatory cytokines in a severe burn mouse model, which was suppressed by stimulation of SRIT1 levels in response to RSV [48].

In addition to the secretion of pro-inflammatory factors, macrophages are the first line of defence against endocytic bacteria. Pterostilbene, as a methoxylated derivative of RSV, is rapidly phagocytosed by macrophages and facilitating intracellular eradication of methicillin-resistant *S. aureus* [49]. Paul Yao et al. [50] found that pterostilbene can reverse the epigenetic changes of macrophages and inhibit the expression of proinflammatory factors by comparison with RSV in diabetic wound healing, thus accelerating wound healing. In a word, macrophages have an irreplaceable role in wound healing. The use of RSV also impedes macrophages infiltration and the secretion of pro-inflammatory factors, which is beneficial in expediting wound healing.

### 3.4. Mesenchymal Stem Cells

Mesenchymal stem cells (MSCs) are pluripotent cells derived from a variety of tissues, including bone marrow, adipose, dental pulp, placenta, umbilical cord and liver. They have a high proliferative and differentiation potential and low immunogenicity, which can be used in the study of a wide range of clinical diseases. MSCs play an important role in cell growth and maintenance of skin tissue through the secretion of a number of cytokines and growth factors. Recent studies have shown that MSCs stimulate human dermal fibroblasts (HDFs) through paracrine action to accelerate wound healing via reducing collagen deposition and scar proliferation [51,52]. In MSCs, RSV dose-dependently promotes the increase of secretory factors such as TGF-β, VEGF, PEGF and EGF and the proliferation of MSCs, contributing to faster wound healing [53]. In a rat model of type 1 diabetes mellitus [54], RSV-stimulated extracellular vesicles secreted by MSCs also contributed to improving diabetic wound healing.

## 4. Mechanisms Underlying the Beneficial Effects of RSV on Wound Healing

### 4.1. Anti-Inflammatory Effect

Inflammation is a physiological stress response that occurs when the organism is exposed to external aggression and can be divided into acute and chronic inflammation. Normally, inflammation is beneficial and is the body’s automatic defence response, but it can be damaging to the organism if it becomes chronic as the disease progresses. Anti-inflammatory effects therefore play a vital role in the fight against pathogens and wound healing. Macrophages are the most predominant type of immune cells in which the inflammatory response of the body occurs. In vitro experiments, macrophages are usually induced to produce an inflammatory response by lipopolysaccharide. Apparently, RSV in its natural form or in wound dressings inhibits the production of certain pro-inflammatory cytokines released into the cell supernatants, such as COX-2, and its derivatives PGE2, TNF-α and IL-1β decrease with increasing duration of RSV action [47], or down-regulate the expression of the corresponding genes in cells [55,56]. In a rat burn wound model, RSV-loaded hydrogels inhibited the expression of inflammatory factors such as IL-6, IL-1β and TNF-α [21].

The anti-inflammatory effects of RSV in wound healing are closely linked to SIRT1 [57]. Several studies have found that RSV reduces the expression of APE1/Ref-1, a common target of SIRT1, in lipopolysaccharide-stimulated monocytes and shifts it from the cytoplasm to the nucleus, suggesting that acetylation of proteins occurs under the action of RSV. There is accumulated evidence to show [47] that RSV can reduce excessive collagen formation by activating SIRT-1, thereby reducing scar formation and accelerating skin wound healing. In addition, SIRT1 mediated endothelial protection and pro-angiogenic effects in the RSV-treated mouse diabetic wound model [58]. Although it is still not fully understood how RSV exerts its biological functions, its activation of SIRT1 seems to have been considered a necessary link.

Nuclear factor kappa b (NF-ĸB) has been validated as a common anti-inflammatory mechanism in a variety of diseases, including respiratory diseases [59], enteropathy [60], mechanical abnormal pain [61], and others. Certainly, NF-κB was also found to be involved in the anti-inflammatory role of RSV during skin injury by James E Sligh et al. [62]. In RAW 264.7 cells [63], the treated group of biopolymer nanoparticles encapsulated with RSV promoted the release of the anti-inflammatory factor IL-10 and inhibited the expression of NO, IL-1β, IL-6, in which MAPK phosphorylation was inhibited, suggesting that RSV can mediate its anti-inflammatory mechanisms through the MAPK signalling pathway. Interestingly, some novel ideas [64] suggest that the anti-inflammatory and wound healing effects of RSV depend on their interaction with Epidermal growth factor receptor (EGFR)-controlled cytoplasmic and nuclear pathways rather than on their direct redox properties.

### 4.2. Anti-Microbial Effect

Avoiding local wound infections in the early stages of tissue damage is an essential part of early treatment. Most common clinical infections are bacterial and fungal, with *S. aureus*, *P. aeruginosa* and *C. albicans* predominating [65]. Therefore, the use of antimicrobial agents is essential in the early treatment of the tissue repair process. In modern clinical medicine, the widespread use of antibiotics has made resistance to microbial pathogens a major clinical problem. The search for an effective drug with fewer side effects that can resist/resolve this disadvantage of drug resistance has therefore become a new direction. RSV, a natural plant extract, has been shown to have antimicrobial properties and its antimicrobial activity may be due to disruption of cell membrane potential or blockage of DNA synthesis. In one study, RSV was found to be antibacterial against both bacterial and fungal infections [66]. However, compared to *S. aureus* and *P. aeruginosa*, RSV did not hinder the growth of *C. albicans* although it also inhibited it better, which means that the inflammatory response is stronger in fungal infections than in bacterial infections. Furthermore, as a natural plant extract, RSV has a better antibacterial effect compared to other extracts. Three plant polyphenols containing RSV, dihydroquercetin and dihydromyricetin have been extracted from the bark of the fir tree. Compared to the latter two, RSV treatment resulted in a significant increase in the proportion of macrophages and fibroblasts, which became important in accelerating wound healing and restoring the normal stratification of the epidermis and hair follicles [67]. In addition, RSV had been found to be an effective compound against *Stenotrophomonas* in burn wounds [68]. It is thus clear that RSV could be a candidate for antimicrobial therapy.

### 4.3. Antioxidant Effect

Oxidative stress and inflammation go hand in hand. Therefore, having oxidative stress is also an important factor in wound healing. In general, oxidative stress usually is a result of excessive reactive oxygen species (ROS), which might be the important mechanism of delayed wound healing [69]. RSV is a natural antioxidant with powerful antioxidant effects against damaging diseases. In H_2_O_2_-induced oxidative stress in Human Umbilical Vein Endothelial Cells (HUVECs), RSV was found to inhibit ROS production by promoting Nrf2 aggregation in the nucleus and stimulating the expression of the downstream factor Mn-SOD [70]. Similarly, the results were consistently validated in a rat model of skin burn trauma. However, it has also been found that the antioxidant properties of RSV may have a biphasic concentration effect. The effect of 0.1uM RSV reduced JNK and COX-2 expression in keratinocytes and upregulated expression at 1uM provides evidence for this theory [41].

### 4.4. Anti-Ageing Effect

With age comes an increasing risk of chronic non-healing skin wounds. It has been shown that RSV has anti-ageing effects during tissue damage repair [71]. The anti-aging effect of RSV is dependent on continuous dosing, and animal studies have demonstrated that intermittent dosing does not improve wound healing in young rats. In aged rats, continuous RSV treatment down-regulated senescence factors such as p16, p21, promoted cell proliferation in the epidermis and perifollicular areas, and improved senescence in skin wound healing, confirming the anti-aging effect of RSV in wound healing [72]. Moreover, CD31 mapping in granulation tissue confirmed that RSV promotes the increase of CD31^+^ vascular endothelial cells by stimulating adenosine monophosphate-activated protein kinase (AMPK), thereby accelerating angiogenesis and wound healing. In addition, the anti-ageing activity of RSV in combination with other phytonutrients has been reported in the context of skin damage [73]. The following mapping illustrates the function of RSV in wound healing and its possible mechanisms (Figure 3).

The role of RSV in wound healing is chiefly manifested in four aspects, which contains anti-oxidation, anti-inflammatory, anti-aging and anti-bacterial effects, and its brief mechanism of action is elucidated.

## 5. Cytotoxicity and Adverse Effects of RSV on Wound Healing

The cytotoxicity potential of RSV resisting human wound healing demonstrated that RSV had low associated cytotoxicity. Most studies have shown that higher doses of RSV lead to cell proliferation inhibition and cell viability decline, such as T cells, macrophages, skin cells and mammalian cells [74,75]. In contrast, the bioeffects of low dose (10 µm/mL) RSV did not affect its ability for cell morphology and function [76]. In distinction, a case for macrophages, the degree of cytotoxic appears to be related to the activation status of the cells. The investigators also found that the elevated glutathione and stable levels of lipid peroxidation suggest that RSV exhibited cytotoxicity that did not involve oxidative stress [77]. In conclusion, the current cytotoxicity assay indicates that this compound has low cytotoxicity and a preliminary safety profile.

## 6. Discussion

In recent years, there has been an increasing interest in the healing of skin wounds. The healing of some mechanical injuries, burns, diabetic wounds and others has become a clinical challenge and the resistance to antibiotics in modern medicine has become a therapeutic bottleneck. As a result, natural plant extracts have become the target of focus. RSV is found in a variety of plants, including beans, rice and grape skins. It has also been found to have anti-inflammatory, antioxidant, antibacterial and even pro-angiogenic effects in a variety of diseases, of course, containing the healing of skin wounds. Due to the high gastrointestinal degradation and low bioavailability of RSV orally, several forms of dressings have been investigated and developed. In this review, we focussed on the effects of different forms of RSV-loaded dressings on the proliferation and migration of different cells during skin wound healing and also elaborated on their anti-inflammatory, antioxidant and antimicrobial properties and mechanisms of action in the treatment of wound healing. Several experiments have demonstrated its beneficial effects on wound healing at the cellular level and in animal studies. However, in reviewing the literature, we found that the dose-dependence of RSV drugs is particularly pronounced, with high or low concentrations of action even leading to contradictory results, which offers us a new approach to the treatment of different diseases. Considering that RSV promotes wound healing by activating fibroblast proliferation or apoptosis [32,78], it reminds us to wonder whether greater attention should be paid to reducing scar formation in the process of accelerating wound healing. While it is indisputable that fibroblasts proliferation accelerates wound healing, several studies [79,80] have shown that fibroblasts proliferation during the granulation phase and tissue remodelling leads to the deposition of collagen fibres and ECM, resulting in the formation of scarring, which has a significant impact on the physiological function of the skin and people’s psychological well-being. Therefore, more in-depth studies are needed on the treatment of fibroblasts proliferation by RSV and its phase of action.

## Figures and Tables

**Figure 1 molecules-27-06736-f001:**
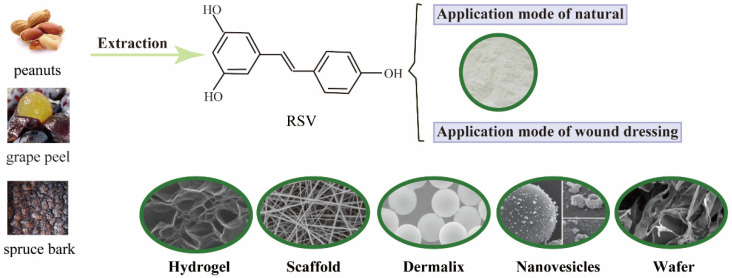
The native and wound dressing modes of RSV.

**Figure 2 molecules-27-06736-f002:**
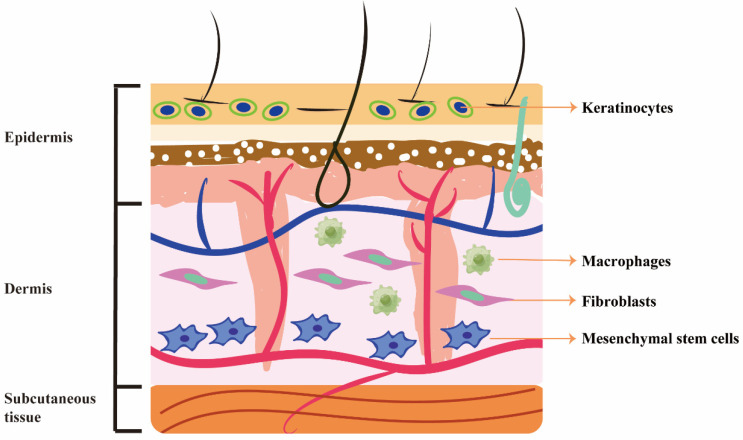
Beneficial cells of RSV in wound healing.

**Figure 3 molecules-27-06736-f003:**
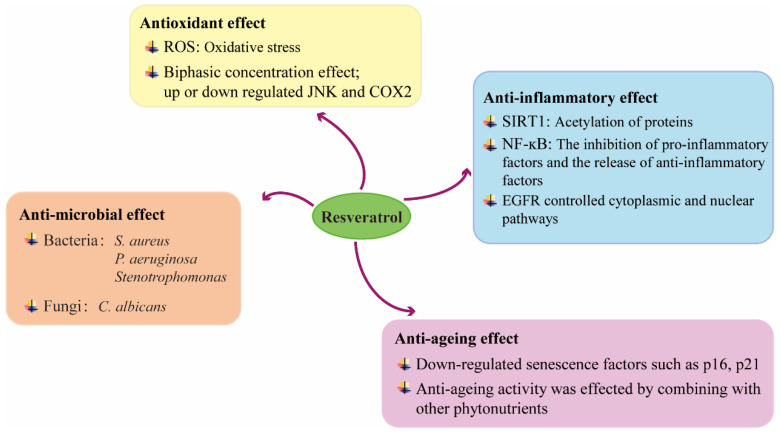
Mechanisms underlying the beneficial effects of RSV on wound healing.

## Data Availability

Not applicable.

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
