# Peer review of "Emerging Effects of Resveratrol on Wound Healing: A Comprehensive Review"

_molecules, 2022, doi:10.3390/molecules27196736_

Round 1

Reviewer 1 Report

Manuscript iD: molecules-1930679

Authors: Jia et al.,

In this review article entitled “Emerging effects of resveratrol on wound healing: a comprehensive review”, the authors focused on summarizing the beneficial effect of resveratrol and its dressings’ applications on wound healing

Hereafter, some points that should be taken into account before acceptance and publication.

Comments to the authors:

-          In the section 3. Beneficial effects of RSV at different cells, the authors should include the relationship of the beneficial effect on the different damaged skin layers in the wound healing process.

-          The 2D chemical structure that is provided in figure 1 should be labeled. Idem for the photos given in the left side of the figure.

-          The beneficial effects of RSV on both macrophages and microphages during wound healing is overlooked.

-          Figure 2 contains a regular title and legend at the bottom but also an extra title is provided and incorporated in the top of the figure itself. The one on the top should be deleted.

-          The sentence in lines 288-290 : “In general, wound healing….” should be revised as oxidative stress usually is a result of excessive reactive oxygen species (ROS).

-          In figure 3, i) scientific names both bacteria and fungi should be in italic. ii) standardize the way of writing the names: either S. aureus and C. albicans or full names for all.

-          English language is fine but minor checking is required such as:

“effects, which is” in line 15 should be “effects, which are”,

“solution” in line 78 should be “ solutions”,

“on” instead of “at” in line 133,

“inflametory” and “icombination” in figure 3.

Minor comment

-          Spaces should be revised, specifically just before the reference citations except “[6]” in line 113.

-           

Author Response

Dear reviewer,

Thanks for your positive and constructive comments and suggestions. And we have made change in a revised manuscripts.

Best wishes,

Feng-lai Yuan

  • In the section 3. Beneficial effects of RSV at different cells, the authors should include the relationship of the beneficial effect on the different damaged skin layers in the wound healing process.

Response: We add the relationship between skin layers and various cell in the section 3, it is described as follows: “Skin is the largest defence organ against physical, mechanical, chemical and pathogenic microbial attacks and has three main layers: the epidermis, the dermis and the subcutaneous tissue. The keratinocytes are the most important cellular component of the epidermis and produce keratin, which acts as a barrier to the skin. Fibroblasts, macrophages and mesenchymal stem cells are mainly located in the dermis and are recruited to the site of injury and promote collagen formation and contraction during trauma, thus contributing to the healing of the wound.”

-  The 2D chemical structure that is provided in figure 1 should be labeled. Idem for the photos given in the left side of the figure.

Response: We labeled the 2D chemical structure in figure 1.

  • The beneficial effects of RSV on both macrophages and microphages during wound healing is overlooked.

Response: Thank you for your suggestion. The content of macrophages is as follows:

3.3 Macrophages

Macrophages are important inflammatory cells that play a key role in fighting infection in the body during wound healing. However, it has been suggested that an increase in excessive inflammatory cytokines is positively associated with the severity of trauma [26]. Therefore, balancing macrophage homeostasis is also a critical step in wound healing. During the early stages of wound healing, macrophages produce several pro-inflammatory cytokines as IL-6, IL-1b and tumour necrosis factor-α (TNF-α) [27]. RSV-loaded hydrogels reduced macrophage infiltration and the expression of cytokines such as TNF-α and IL-1b in rat skin damage model, which promotes shrinkage and repair of wounds [28]. Similarly, lipopolysaccharide-induced macrophage activation exhibited a marked increase in inflammatory cytokines in a severe burn mouse model, which was suppressed by stimulation of SRIT1 levels in response to RSV [29].

In addition to the secretion of pro-inflammatory factors, macrophages are the first line of defence against endocytic bacteria. Pterostilbene, as a methoxylated derivative of RSV, is rapidly phagocytosed by macrophages and facilitating intracellular eradication of methicillin-resistant S. aureus [30]. Paul Yao et al. [31] found that pterostilbene can reverse the epigenetic changes of macrophages and inhibit the expression of proinflammatory factors by comparison with RSV in diabetic wound healing, thus accelerating wound healing. In a word, macrophages have an irreplaceable role in wound healing. And the use of RSV impedes macrophage infiltration and the secretion of pro-inflammatory factors, which is beneficial in expediting wound healing.

  • Figure 2 contains a regular title and legend at the bottom but also an extra title is provided and incorporated in the top of the figure itself. The one on the top should be deleted.

Response: The title on the top of Figure 2 and Figure 3 have been deleted.

  • The sentence in lines 288-290 : “In general, wound healing….” should be revised as oxidative stress usually is a result of excessive reactive oxygen species (ROS).

Response: This sentence has been revised as “ In general, oxidative stress usually is a result of excessive reactive oxygen species (ROS), which might be the important mechanism of delayed wound healing ”.

  • In figure 3, i) scientific names both bacteria and fungi should be in italic. ii) standardize the way of writing the names: either  aureus and C. albicans or full names for all.

Response: Scientific names both bacteria and fungi in the text and Figure 3 have been revised to be consistent.

  • English language is fine but minor checking is required such as: “effects, which is” in line 15 should be “effects, which are”,

Response: This sentence has been revised as “ …effects, which are closely associated with a variety of injurious diseases and even cosmetic medicine ”.

  • “solution” in line 78 should be “ solutions”,

Response: Thank you for your suggestion, which has been incorporated in the revised version.

  • “on” instead of “at” in line 133,

Response: Thank you for your suggestion, which has been incorporated in the revised version.

  • “inflametory” and “icombination” in figure 3.

Response: The mistake has been corrected.

Minor comment

-          Spaces should be revised, specifically just before the reference citations except “[6]” in line 113.

Response: Thank you for your suggestion, which has been changed in the revised version.

Reviewer 2 Report

The present review paper is an overview of the resveratrol (RSV) involvement in wound healing. Up to date, there is no review collecting the specific data for the resveratrol role in wound healing covering the addressed, which are: wound dressing-containing resveratrol, beneficial effects of RSV on different cellular types, mechanisms underlying the beneficial effects of RSV on wound healing (anti-inflammatory effect, anti-microbial effect, antioxidant effect). 

The review is clear and comprehensive with relevance in the field. The gap in the knowledge was identified. There is no similar review published recently, so the present one is relevant and of interest for the scientific community.

The references are mostly recent (within the last 5 years) and relevant, but you should consider to include some other references (see one very recent reference below, just as example). There are no self-citations listed in the references.

E.g.: Hecker A, Schellnegger M, Hofmann E, Luze H, Nischwitz, Kamolz LP, Kotzbeck P. The impact of resveratrol on skin wound healing, scarring, and aging. International Wound Journal, 2022; 19(1): 9-28

The statements and conclusions are coherent and supported by the listed citations.

The review includes three original figures, with a short title. I would recommend a more extensive explanation of the figure’s components in the caption section (e.g: line 65, Figure 1 – The RVS can be applied as natural compound or within different wound dressings such as hydrogel, …etc.). The same should be done for Figure 2 (line 142) and Figure 3 (line 314).

Author Response

Dear reviewer,

Thanks for your positive and constructive comments and suggestions. And we have made change in the follow pages.

Best wishes,

Feng-lai Yuan

The present review paper is an overview of the resveratrol (RSV) involvement in wound healing. Up to date, there is no review collecting the specific data for the resveratrol role in wound healing covering the addressed, which are: wound dressing-containing resveratrol, beneficial effects of RSV on different cellular types, mechanisms underlying the beneficial effects of RSV on wound healing (anti-inflammatory effect, anti-microbial effect, antioxidant effect). 

The review is clear and comprehensive with relevance in the field. The gap in the knowledge was identified. There is no similar review published recently, so the present one is relevant and of interest for the scientific community.

  • The references are mostly recent (within the last 5 years) and relevant, but you should consider to include some other references (see one very recent reference below, just as example). There are no self-citations listed in the references.

E.g.: Hecker A, Schellnegger M, Hofmann E, Luze H, Nischwitz, Kamolz LP, Kotzbeck P. The impact of resveratrol on skin wound healing, scarring, and aging. International Wound Journal, 2022; 19(1): 9-28

Response: Thank you for the suggestion. We have searched the relevant reviews on the role of resveratrol in wound healing in the past two years and have added it in the references.

  1. Pignet, A.L.; Schellnegger, M.; Hecker, A.; Kohlhauser, M.; Kotzbeck, P.; Kamolz, L.P. Resveratrol-Induced Signal Transduction in Wound Healing. International journal of molecular sciences 2021, 22, doi:10.3390/ijms222312614.
  2. Hecker, A.; Schellnegger, M.; Hofmann, E.; Luze, H.; Nischwitz, S.P.; Kamolz, L.P.; Kotzbeck, P. The impact of resveratrol on skin wound healing, scarring, and aging. International wound journal 2022, 19, 9-28, doi:10.1111/iwj.13601.
  • The review includes three original figures, with a short title. I would recommend a more extensive explanation of the figure’s components in the caption section (e.g: line 65, Figure 1 – The RVS can be applied as natural compound or within different wound dressings such as hydrogel, …etc.). The same should be done for Figure 2 (line 142) and Figure 3 (line 314).

Response: A more extensive explanation of the figure’s components in the caption section have been supplemented.

Figure 1. The native and wound dressing modes of RSV

Resveratrol is a natural compound derived from peanut, grape peel and spruce bark. Currently, the application of resveratrol is divided into natural compound and loaded dressing, which includes hydrogels, scaffolds, Dermalix, nanovesicles and wafer.

Figure 2. Beneficial cells of RSV in wound healing

Skin is mainly divided into epidermis, dermis and subcutaneous tissue. There are many cells involved in the wound healing process, including keratinocytes, fibroblasts, macrophages, and mesenchymal stem cells.

Figure 3 Mechanisms underlying the beneficial effects of RSV on wound healing.

The role of RSV in wound healing is chiefly manifested in four aspects, which contains anti-oxidation, anti-inflammatory, anti-aging and anti-bacterial effects. And its brief mechanism of action is elucidated.

Reviewer 3 Report

The review titled “Emerging effects of resveratrol on wound healing: a comprehensive review

” and authored by Dr Jia et al. summarizes the role of resvera-16 trol and its different forms of action in wound healing. The manuscript worth publication in Molecules after Major revision after considering the following:

The whole manuscript lake of references citations. For instance: in the introduction 1st paragraph line 26-34 has no references at all and 2n paragraph line 36-49 has only one reference!!

A description of the topics will be covered should be outlined at the end of the introduction.

Author is advised to include a separated section with the adverse effect of RSV including its toxicity and complications associated with RSV application 60 in wound healing.

Author Response

Dear reviewer,

Thanks for your positive and constructive comments and suggestions. And we have made change in the follow pages.

Best wishes,

Feng-lai Yuan

  • The whole manuscript lake of references citations. For instance: in the introduction 1st paragraph line 26-34 has no references at all and 2n paragraph line 36-49 has only one reference!!

Response: Thank you for the suggestion. We have added some new references in the introduction as follows:

  1. Bryan, D.; Walker, K.B.; Ferguson, M.; Thorpe, R. Cytokine gene expression in a murine wound healing model. Cytokine 2005, 31, 429-438, doi:10.1016/j.cyto.2005.06.015.
  2. Materazzi, S.; Pellerito, S.; Di Serio, C.; Paglierani, M.; Naldini, A.; Ardinghi, C.; Carraro, F.; Geppetti, P.; Cirino, G.; Santucci, M.; et al. Analysis of protease-activated receptor-1 and -2 in human scar formation. The Journal of pathology 2007, 212, 440-449, doi:10.1002/path.2197.
  3. Li, C.; Wei, S.; Xu, Q.; Sun, Y.; Ning, X.; Wang, Z. Application of ADSCs and their Exosomes in Scar Prevention. Stem cell reviews and reports 2022, 18, 952-967, doi:10.1007/s12015-021-10252-5.
  4. Gao, Y.; Hou, X.; Dai, Y.; Yang, T.; Chen, K. Radiation-induced FAP + fibroblasts are involved in keloid recurrence after radiotherapy. Frontiers in cell and developmental biology 2022, 10, 957363, doi:10.3389/fcell.2022.957363.
  5. Valletta, A.; Iozia, L.M.; Leonelli, F. Impact of Environmental Factors on Stilbene Biosynthesis. Plants (Basel, Switzerland) 2021, 10, doi:10.3390/plants10010090.
  6. Bryl, A.; Falkowski, M.; Zorena, K.; Mrugacz, M. The Role of Resveratrol in Eye Diseases-A Review of the Literature. Nutrients 2022, 14, doi:10.3390/nu14142974
  • A description of the topics will be covered should be outlined at the end of the introduction.

Response: Thank you for the suggestion. We have added the description of topics in the end of introduction, which is as follows: "We described the effects of RSV on different effector cells in the process of wound healing. And the underlying mechanisms of RSV action were summarized, containing anti-inflammatory, antibacterial, antioxidant and anti-aging effects. "

  • Author is advised to include a separated section with the adverse effect of RSV including its toxicity and complications associated with RSV application 60 in wound healing.

Response: Thank you for your suggestion. The toxicity of RSV is as follows:

  1. Cytotoxicity and adverse effects of RSV on wound healing

The use of natural compounds cannot be separated from cytotoxicity tests. Most studies [51,52] have shown that higher doses of RSV lead to cell prolifation inhibition and cell viability decline, such as T cells, macrophages, skin cells and mammalian cells. But low dose RSV did not affect cell morphology and function. Distinguishingly, a case for macrophages, the degree of cytotoxic appears to be related to the activation status of the cells. The investigators also found that the elevated glutathione and stable levels of lipid peroxidation suggest that RSV exhibited cytotoxicity that did not involve oxidative stress [53]. In conclusion, the current cytotoxicity assay indicates that this compound has low cytotoxicity and a preliminary safety profile.

Round 2

Reviewer 1 Report

ROUND #2

Manuscript molecules-1930679

After providing a revised version along with a response to the reviewers’ comments, we can obviously notice that the manuscript has been sufficiently improved to warrant publication in molecules. Just minor modifications are still needed.

1-      “ROS” (In figure 3), stands for “Reactive Oxygen Species” not for ”Oxidative stress”.

2-      Illustrated macrophages in figure 2, should have typical nuclei. The nucleus should have kidney-shaped (or coffee bean-shaped structure).

Author Response

Dear Reviewer:

Thank you for your suggestions and encouragement.  We have made changes in a revised manuscript (highlighted in red), which we hope you will reconsider for publication.

With best wishes,

Feng Lai Yuan

  • “ROS” (In figure 3), stands for “Reactive Oxygen Species” not for ”Oxidative stress”.

         Response: Thank you for the suggestion. Figure 3 has shown the relationship between ROS and Oxidative stress in antioxidant effect of RSV during wound healing.

  •  Illustrated macrophages in figure 2, should have typical nuclei. The nucleus should have kidney-shaped (or coffee bean-shaped structure).

         Response: Thank you for the suggestion. We changed the morphology of macrophages in Figure 2.

Reviewer 3 Report

Thanks for the revised version; however I do not see that the authors have taken comments seriously and my concerns still the same.

Even the author added four references (not six as mentioned in the revision) to the introdcution, they are not enough and still whole sections lack citations for example "The performance forms of RSV to Wound Healing" and even references are not distrubuted equally.

Line 49 starting with And (should be and) please revise the whole manuscript . for typos and grammatical errors.

The newly added cytotoxicity section"Cytotoxicity and adverse effects of RSV on wound healing" is written in bad english. What does it means "The use of natural compounds cannot be separated from cytotoxicity tests". Also, references should be inserted at the end of pragraph (e.g., 51,52).

What is the safe dose of RSV for cells (give concentrations). Indeed, this section lakes proper citations.

Author Response

Dear Reviewer:

Thank you for your suggestions and encouragement.  We have made changes in a revised manuscript (highlighted in red), which we hope you will reconsider for publication.

With best wishes,

Feng Lai Yuan

  • Even the author added four references (not six as mentioned in the revision) to the introdcution, they are not enough and still whole sections lack citations for example "The performance forms of RSV to Wound Healing" and even references are not distrubuted equally.

      Response: Thank you for the suggestion. We have added some new references in the introduction and whole manuscripts as follows:

     1. Butkeviciute, A.; Ramanauskiene, K.; Kurapkiene, V.; Janulis, V. Dermal Penetration Studies of Potential Phenolic Compounds Ex Vivo and Their Antioxidant Activity In Vitro. Plants (Basel, Switzerland) 2022, 11, doi:10.3390/plants11151901.

     3. Eroğlu, İ.; Gökçe, E.H.; Tsapis, N.; Tanrıverdi, S.T.; Gökçe, G.; Fattal, E.; Özer, Ö. Evaluation of characteristics and in vitro antioxidant properties of RSV loaded hyaluronic acid-DPPC microparticles as a wound healing system. Colloids and surfaces. B, Biointerfaces 2015, 126, 50-57, doi:10.1016/j.colsurfb.2014.12.006.

     4. Li, H.; O'Meara, M.; Zhang, X.; Zhang, K.; Seyoum, B.; Yi, Z.; Kaufman, R.J.; Monks, T.J.; Wang, J.M. Ameliorating Methylglyoxal-Induced Progenitor Cell Dysfunction for Tissue Repair in Diabetes. Diabetes 2019, 68, 1287-1302, doi:10.2337/db18-0933.

     5. Huang, X.; Sun, J.; Chen, G.; Niu, C.; Wang, Y.; Zhao, C.; Sun, J.; Huang, H.; Huang, S.; Liang, Y.; et al. Resveratrol Promotes Diabetic Wound Healing via SIRT1-FOXO1-c-Myc Signaling Pathway-Mediated Angiogenesis. Frontiers in pharmacology 2019, 10, 421, doi:10.3389/fphar.2019.00421.

     9. Vaňková, E.; Paldrychová, M.; Kašparová, P.; Lokočová, K.; Kodeš, Z.; Maťátková, O.; Kolouchová, I.; Masák, J. Natural antioxidant pterostilbene as an effective antibiofilm agent, particularly for gram-positive cocci. World journal of microbiology & biotechnology 2020, 36, 101, doi:10.1007/s11274-020-02876-5.

     10. Tripathi, V.; Chhabria, S.; Jadhav, V.; Bhartiya, D.; Tripathi, A. Stem Cells and Progenitors in Human Peripheral Blood Get Activated by Extremely Active Resveratrol (XAR™). Stem cell reviews and reports 2018, 14, 213-222, doi:10.1007/s12015-017-9784-7.

     11. Ávila-Salas, F.; Marican, A.; Pinochet, S.; Carreño, G.; Valdés, O.; Venegas, B.; Donoso, W.; Cabrera-Barjas, G.; Vijayakumar, S.; Durán-Lara, E.F. Film Dressings Based on Hydrogels: Simultaneous and Sustained-Release of Bioactive Compounds with Wound Healing Properties. Pharmaceutics 2019, 11, doi:10.3390/pharmaceutics11090447.

     12. Valachová, K.; Šoltés, L. Self-Associating Polymers Chitosan and Hyaluronan for Constructing Composite Membranes as Skin-Wound Dressings Carrying Therapeutics. Molecules (Basel, Switzerland) 2021, 26, doi:10.3390/molecules26092535.

      13. Chen, K.; Sivaraj, D.; Davitt, M.F.; Leeolou, M.C.; Henn, D.; Steele, S.R.; Huskins, S.L.; Trotsyuk, A.A.; Kussie, H.C.; Greco, A.H.; et al. Pullulan-Collagen hydrogel wound dressing promotes dermal remodelling and wound healing compared to commercially available collagen dressings. Wound repair and regeneration : official publication of the Wound Healing Society [and] the European Tissue Repair Society 2022, 30, 397-408, doi:10.1111/wrr.13012.

     14. Yang, G.; Zhang, Z.; Liu, K.; Ji, X.; Fatehi, P.; Chen, J. A cellulose nanofibril-reinforced hydrogel with robust mechanical, self-healing, pH-responsive and antibacterial characteristics for wound dressing applications. Journal of nanobiotechnology 2022, 20, 312, doi:10.1186/s12951-022-01523-5.

     15. Poornima, B.; Korrapati, P.S. Fabrication of chitosan-polycaprolactone composite nanofibrous scaffold for simultaneous delivery of ferulic acid and resveratrol. Carbohydrate polymers 2017, 157, 1741-1749, doi:10.1016/j.carbpol.2016.11.056.

     16. Dickmeis, C.; Kauth, L.; Commandeur, U. From infection to healing: The use of plant viruses in bioactive hydrogels. Wiley interdisciplinary reviews. Nanomedicine and nanobiotechnology 2021, 13, e1662, doi:10.1002/wnan.1662.

     17. Boominathan, T.; Sivaramakrishna, A. Recent Advances in the Synthesis, Properties, and Applications of Modified Chitosan Derivatives: Challenges and Opportunities. Topics in current chemistry (Cham) 2021, 379, 19, doi:10.1007/s41061-021-00331-z.

     18. Lan, W.; Zhao, J.; Sun, Y.; Liu, J.; Xie, J. Chitosan-grafted-phenolic acid copolymers against Shewanella putrefaciens by disrupting the permeability of cell membrane. World journal of microbiology & biotechnology 2022, 38, 73, doi:10.1007/s11274-022-03261-0.

     29. Chen, L.; Tredget, E.E.; Wu, P.Y.; Wu, Y. Paracrine factors of mesenchymal stem cells recruit macrophages and endothelial lineage cells and enhance wound healing. PloS one 2008, 3, e1886, doi:10.1371/journal.pone.0001886.

     65. Brown, H.L.; Clayton, A.; Stephens, P. The role of bacterial extracellular vesicles in chronic wound infections: Current knowledge and future challenges. Wound repair and regeneration : official publication of the Wound Healing Society [and] the European Tissue Repair Society 2021, 29, 864-880, doi:10.1111/wrr.12949.

     74. Saleh, H.A.; Ramdan, E.; Elmazar, M.M.; Azzazy, H.M.E.; Abdelnaser, A. Comparing the protective effects of resveratrol, curcumin and sulforaphane against LPS/IFN-γ-mediated inflammation in doxorubicin-treated macrophages. Scientific reports 2021, 11, 545, doi:10.1038/s41598-020-80804-1.

  • Line 49 starting with And (should be and) please revise the whole manuscript . for typos and grammatical errors.

      Response: This sentence has been revised in manuscripts.

  • The newly added cytotoxicity section"Cytotoxicity and adverse effects of RSV on wound healing" is written in bad english. What does it means "The use of natural compounds cannot be separated from cytotoxicity tests". Also, references should be inserted at the end of pragraph (e.g., 51,52).

      Response: We has revised the sentence "The use of natural compounds cannot be separated from cytotoxicity tests" as "Cytotoxicity potential of RSV resist human wound healing demonstrated that RSV had low associated cytotoxicity". In addition, references has been inserted at the end of paragraph.

  • What is the safe dose of RSV for cells (give concentrations). Indeed, this section lakes proper citations.

      Response: We has revised the sentence "But low dose RSV did not affect cell morphology and function" as "In contrast, the bioeffects of low dose (10µm/ml) RSV did not affect its ability of cell morphology and function [59] ".